# Reciprocal analyses in zebrafish and medaka reveal that harnessing the immune response promotes cardiac regeneration

Shih-Lei Lai[1]*, Rubén Marín-Juez[1], Pedro Luís Moura[1†], Carsten Kuenne[2], Jason Kuan Han Lai[1], Ayele Taddese Tsedeke[1], Stefan Guenther[2], Mario Looso[2], Didier YR Stainier[1]*

[1]Department of Developmental Genetics, Max Planck Institute for Heart and Lung Research, Bad Nauheim, Germany; [2]ECCPS Bioinformatics and Deep Sequencing Platform, Max Planck Institute for Heart and Lung Research, Bad Nauheim, Germany

**Abstract** Zebrafish display a distinct ability to regenerate their heart following injury. However, this ability is not shared by another teleost, the medaka. In order to identify cellular and molecular bases for this difference, we performed comparative transcriptomic analyses following cardiac cryoinjury. This comparison points to major differences in immune cell dynamics between these models. Upon closer examination, we observed delayed and reduced macrophage recruitment in medaka, along with delayed neutrophil clearance. To investigate the role of immune responses in cardiac regeneration, we delayed macrophage recruitment in zebrafish and observed compromised neovascularization, neutrophil clearance, cardiomyocyte proliferation and scar resolution. In contrast, stimulating Toll-like receptor signaling in medaka enhanced immune cell dynamics and promoted neovascularization, neutrophil clearance, cardiomyocyte proliferation and scar resolution. Altogether, these data provide further insight into the complex role of the immune response during regeneration, and serve as a platform to identify and test additional regulators of cardiac repair.

**\*For correspondence:** ben.s.lai@gmail.com (S-LL); Didier.Stainier@mpi-bn.mpg.de (DYRS)

**Present address:** [†]School of Biochemistry, University of Bristol, Bristol, United Kingdom

## Introduction

Heart failure results from the inability of the human heart to replenish damaged tissue after myocardial infarction (MI), leading to an unresolved scar, tissue remodeling, and functional impairment. Unlike humans, many vertebrates, including certain fish and amphibians, are able to regenerate their heart (*Vivien et al., 2016*). Comparative analyses have been performed in many species to identify factors that may promote cardiac regeneration in humans (*Vivien et al., 2016*). These studies have revealed that regenerative capacity exists sparsely across the animal kingdom, and seems to correlate with low-metabolic state, poikilothermia, hypoxia, an immature cardiomyocyte (CM) structure, and an immature immune system (*Vivien et al., 2016*). As an example, zebrafish exhibit a remarkable regenerative capacity after different cardiac insults (*Poss et al., 2002*; *González-Rosa et al., 2011*; *Schnabel et al., 2011*; *Chablais et al., 2011*, *Zhang et al., 2013*). Upon cardiac injury, the zebrafish heart exhibits in turn, gene activation in epicardium and endocardium, neovascularization, proliferation of CMs, functional integration of new muscle cells, and replacement of scar tissue (*Kikuchi and Poss, 2012*). However, the profound physiological differences between regenerative and non-regenerative model organisms often mask the informative data directly relevant to

regeneration. Recently, another fresh water teleost, medaka (*Oryzias latipes*), has been reported to lack neovascularization or CM proliferation post cardiac injury, subsequently displaying excessive fibrosis and an unresolved scar (*Ito et al., 2014*). Zebrafish and medaka share a living environment, similar physiological conditions, and anatomical structures (*Furutani-Seiki and Wittbrodt, 2004*). In addition, orthologous genes between these two teleost species are easier to identify compared to more phylogenetically distant organisms, rendering them more suitable for direct comparisons (*Furutani-Seiki and Wittbrodt, 2004*).

To identify factors that account for the differences in cardiac regenerative capacity between zebrafish and medaka, we performed detailed comparative transcriptomic analyses following cardiac injury. From these analyses, we observed that acute immune responses appear to be different between zebrafish and medaka. Tissue inflammation and immune cell recruitment/infiltration occur immediately following injury, and have been shown to be required for heart regeneration in zebrafish (*Huang et al., 2013a*; *de Preux Charles et al., 2016a*) and neonatal mice (*Lavine et al., 2014*; *Aurora et al., 2014*). In addition, the type of the immune response and profile of the immune cells recruited to the injury site can influence the extent of injury repair (*Godwin et al., 2017*). However, the permissive and instructive roles of the various immune cell populations during heart regeneration remain largely unknown.

Consistent with the information from our transcriptomic analyses, we observed delayed and reduced macrophage recruitment in medaka compared to zebrafish following cardiac injury. Delaying macrophage recruitment in the normally regenerative zebrafish resulted in defective neovascularization and neutrophil clearance, decreased CM proliferation and unresolved scar. In contrast, injection of the Toll-like receptor agonist poly I:C into the non-regenerative medaka accelerated immune cell dynamics and promoted neovascularization, CM proliferation, and scar resolution. Altogether, these results highlight the role of an acute immune response and timely macrophage recruitment as key triggers of the cardiac regenerative machinery.

## Results

### Comparative transcriptomic analyses between zebrafish and medaka hearts post injury

To perform comparative transcriptomic analyses, we collected zebrafish and medaka ventricles at several time points following sham operation, cryoinjury, or without any operation (untouched), and pooled 4 ventricles per time point for RNA extraction (*Figure 1A*). We determined the transcriptomes of these samples by RNA sequencing, mapped the reads to their respective genomes, and generated normalized read counts at the gene level. Further, to allow for comparative analyses, we mapped genes orthologous between zebrafish and medaka based on homology.

Hierarchical clustering of 5312 differentially expressed genes (DEGs) provided a global overview of gene expression changes following cardiac injury, showing that the overall responses in zebrafish and medaka are similar (*Figure 1B*). However, distinct clusters where gene expression changes between the two species are highly dissimilar can be identified (marked in *Figure 1B*). For example, clusters A1 and A2 include genes that are highly upregulated in zebrafish following injury, but display an attenuated response in medaka, while clusters B1 and B2 include genes that are highly upregulated following injury in medaka but not in zebrafish. In addition, we noticed more dynamic changes in gene expression in zebrafish at 6 hr post cryoinjury (hpci), suggesting that acute gene responses are more frequent in zebrafish when compared to medaka (*Figure 1B*).

To visualize the major trends in gene expression changes, we conducted Principal Component Analysis (PCA) based on normalized counts of all genes expressed in untouched and cryoinjured fish (*Figure 1C*). As expected, the samples were separated according to species in component 1 (*Figure 1C*). The gene expression changes after cryoinjury were temporally similar between zebrafish and medaka in component 2, yet less dynamic in medaka (*Figure 1C*). This finding may imply that the response to cryoinjury in medaka, although muted, is globally comparable to that in zebrafish. Interestingly, while major trends in gene expression in zebrafish reversed back toward the untouched state after 2 days post cryoinjury (dpci), a similar reversion was not observed in medaka until 3 dpci (*Figure 1C*). These results suggest that on a global scale, medaka respond to cardiac injury similarly to zebrafish, but the response in medaka appears to be less dynamic and delayed. To identify major

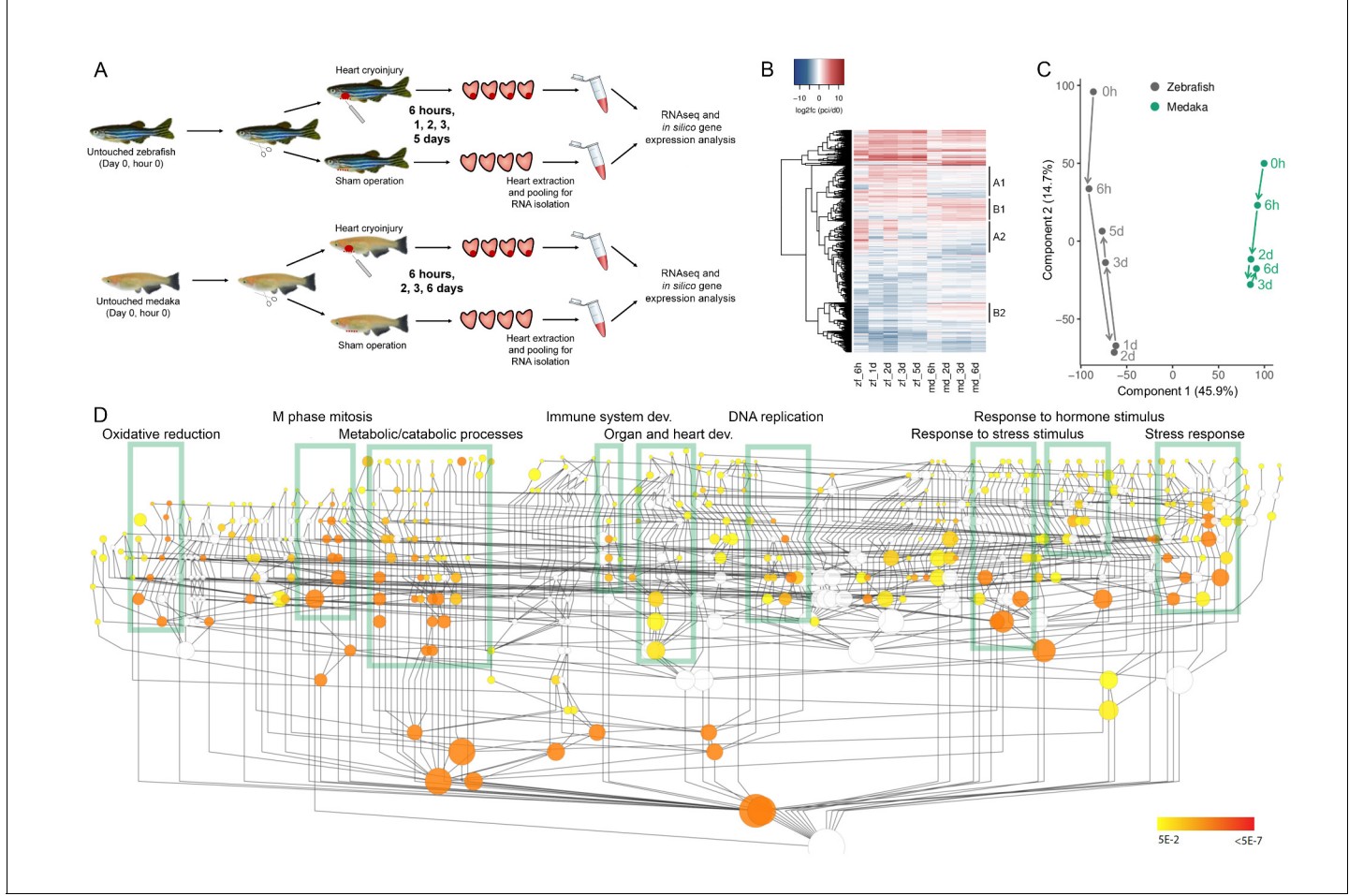

**Figure 1.** Comparative transcriptomic analyses between zebrafish and medaka after cardiac injury. (A) Experimental design. Untouched, sham-operated, and cryoinjured zebrafish and medaka ventricles were collected at 6 hr, 1 d (zebrafish only), 2 d, 3 d, and 5 d (zebrafish only)/6 d (medaka only) post-surgery. Four hearts per time point were pooled, RNA was extracted and subjected to sequencing. (B) Global heatmap depicting expression values of differentially expressed genes vs untouched (0 hr). Hierarchical clustering of genes with an absolute FC of log2 >2 in at least one sample is shown in a heatmap. FC was calculated by comparing to the respective 0 hr expression value. Red color indicates higher expression in the respective sample listed at the bottom of each column, blue color indicates higher expression in the 0 hr sample of the respective organism (medaka or zebrafish). Clusters containing differentially regulated genes between zebrafish and medaka are indicated by A1, A2, B1, and B2. (C) PCA with dimension 1 and 2 of normalized counts. PCA was performed on the normalized counts of untouched and cryoinjured fish, and the first two dimensions are visualized capturing ~60% of the variability of the dataset. (D) Gene Ontology (GO) enrichment of biological processes. Schematic illustration of a GO tree based on biological processes. Each GO term is indicated as a circle whose radius reflects the number of proteins assigned to it. Coloring indicates Benjamini Hochberg corrected p-values<0.05 of hypergeometric tests for overrepresentation/underrepresentation as given in the color scale on the right. The test was performed against the whole annotation reference set. Boxes with respective description mark subtrees of the GO graph that are highly connected and significantly enriched. The given description reflects one of the most enriched terms in the subtree.

responses to cardiac injury in both species, we generated datasets containing relative gene expression levels (log$_2$ fold changes) in cryoinjured hearts compared to untouched hearts, as well as sham-operated hearts compared to untouched hearts, at different time points (*Supplementary file 1*). Using gene ontology (GO) analysis, we found that DEGs were enriched for a number of biological processes including oxidative reduction, M phase and mitosis, metabolic/catabolic processes, immune system development, organ and heart development, DNA replication, response to stress/stimulus, response to hormone stimulus and stress response (*Figure 1D*). These results highlight the major gene expression responses to cardiac injury in zebrafish and medaka.

## Acute inflammation and immune response are blunted in medaka hearts post injury

Next, we investigated differential responses over time between the two species using Sample Level Enrichment Analysis (SLEA). Major GO term categories containing differential responses between zebrafish and medaka include immune response, cell proliferation, angiogenesis, and other biological processes (*Figure 2*, and a complete list in *Supplementary file 2*). Concerning the immune response, zebrafish showed a stronger activation of genes involved in the complement system, macrophages, B cells, T cells, and phagocytosis, while medaka showed a stronger activation of genes related to neutrophil and monocyte chemotaxis (*Figure 2A*). In addition, zebrafish showed a stronger and prolonged activation of genes involved in both cell proliferation (*Figure 2B*) and angiogenesis (*Figure 2C*). Among other biological processes differentially regulated between these two species, we observed stronger activation of genes involved in ROS biosynthesis, hyaluronan metabolism and biosynthesis, blood coagulation, and lipid metabolism in medaka; and a more pronounced activation of genes involved in the response to steroid hormone, adenylate cyclase activity, and adrenergic receptor activity in zebrafish (*Figure 2D*). Interestingly, some of the above-mentioned biological processes, especially the immune response, were also activated in sham-operated hearts compared to untouched hearts (*Figure 2—figure supplement 1*), although the magnitude and

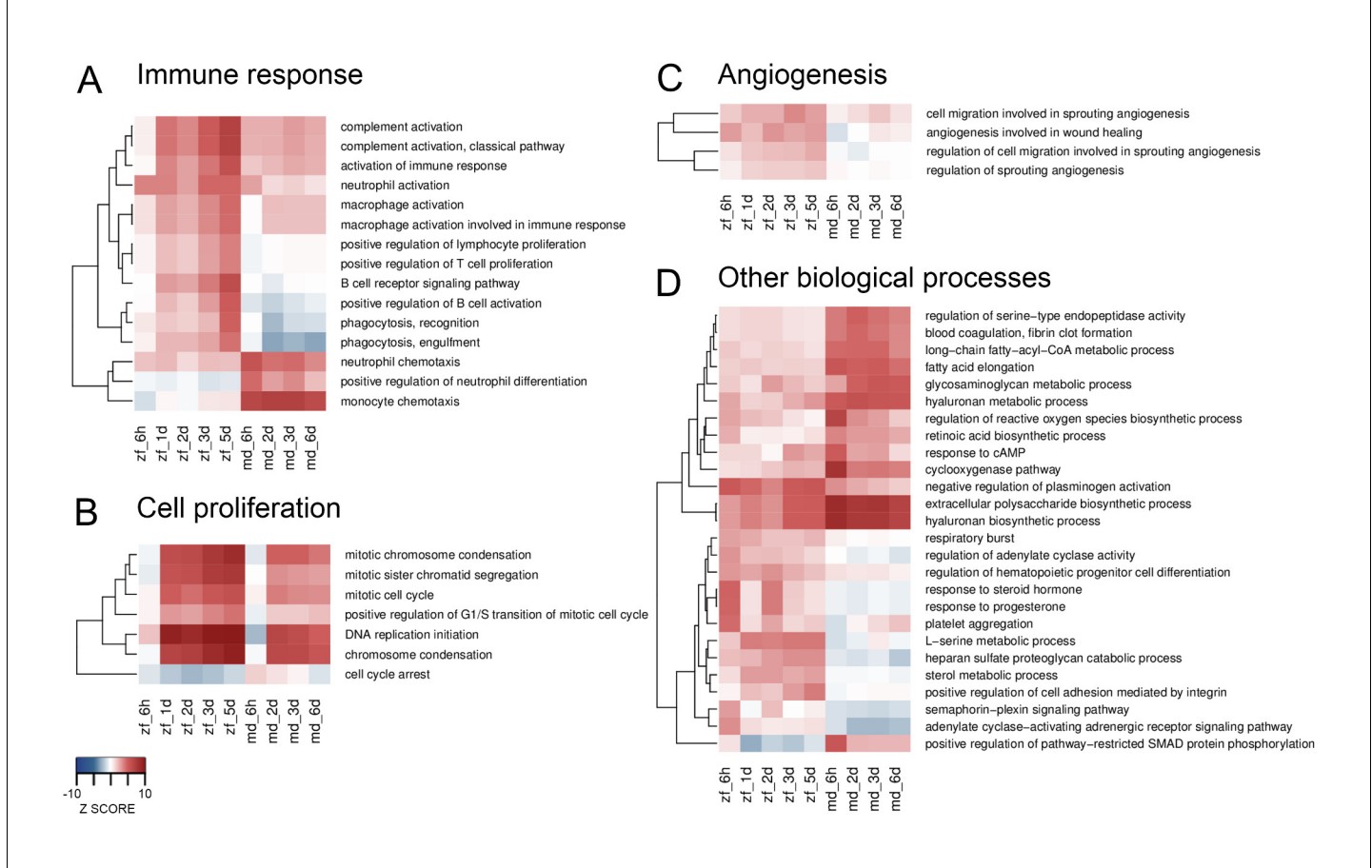

**Figure 2.** Heatmaps of differentially enriched GO terms from SLEA. GO terms from SLEA analysis of cryoinjured vs. untouched hearts related to immune response (A), cell proliferation (B), angiogenesis (C) and other biological processes (D), were selected and respective Z-scores visualized. Color scales were matched for all four heatmaps, ranking from −10 for underrepresentation to +10 for overrepresentation.

The following figure supplement is available for figure 2:

**Figure supplement 1.** Heatmaps of enriched GO terms from SLEA.

dynamics appeared different. For example, genes involved in complement or macrophage activation appear to peak in zebrafish hearts at three days post sham operation, as opposed to a gradual increase over time in injured zebrafish hearts. Accordingly, preconditioning following thoracotomies (similar to our sham operations) has been reported to boost regenerative programmmes in the adult zebrafish heart (*de Preux Charles et al., 2016b*). Comparing our data from sham-operated and untouched hearts further reveals the biological processes potentially involved in this preconditioning (*Figure 2—figure supplement 1*). Altogether, these results show a range of differential responses between zebrafish and medaka following cardiac injury.

To identify canonical pathways and upstream regulators potentially responsible for the differences in cardiac regenerative capacity between the two species, we performed pathway analysis based on genes differentially up or downregulated in zebrafish compared to medaka (top candidates in *Supplementary file 3*, and a full list in *Supplementary file 4*). We identified several immune-related pathways, including phagocytosis, NF-kB, PI3K/AKT, NFAT, and Toll-like receptor signaling (*Supplementary file 3*), that were upregulated in zebrafish. In addition, we found several pathways previously implicated in heart regeneration such as HIPPO (*Heallen et al., 2013*; *Xin et al., 2013*), Neuregulin/ErbB (*Bersell et al., 2009*; *Gemberling et al., 2015*; *D'Uva et al., 2015*), Notch (*Raya et al., 2003*; *Zhao et al., 2014*), and Telomerase activity (*Bednarek et al., 2015*) (*Supplementary file 3*). Analysis of upstream regulators revealed that both poly I:C and CpG ODNs, two TLR agonists, are major upstream regulators of zebrafish responses compared to medaka responses (*Supplementary file 3*). Factors related to TLR signaling including interferon gamma (IFNG) and transforming growth factor beta (TGFB) were also identified (*Supplementary file 3*). TLRs sense various damage-associated molecular patterns (DAMPs) released by damaged tissues, inducing immune responses via activation of downstream genes, including inflammatory cytokines such as IFN, IL1, and TNF, as well as other inflammatory mediators (*Goulopoulou et al., 2016*). Other upstream regulators on the list such as TGFB1 and IGF1 have been reported to positively regulate heart regeneration in zebrafish (*Chablais and Jazwinska, 2012*; *Huang et al., 2013b*), while factors such as RET, MITF, miR-125b and miR101 are potentially interesting candidates for future studies, as their roles in heart regeneration have not yet been examined (*Supplementary file 3*). These results reveal the power of our approach to identify and provide some insight into molecules and pathways whose roles in promoting heart regeneration are largely unknown.

## Macrophage recruitment in medaka heart post injury is delayed and reduced compared to zebrafish

In view of our transcriptomic analyses, we hypothesized that a vigorous and acute immune response is instrumental to promote heart regeneration, while a muted response in medaka may hinder regeneration. Upon tissue damage, innate immune cells detect and respond to DAMPs released at the injury site partly through TLR signaling. Thus, we first examined and compared leukocyte recruitment in zebrafish and medaka hearts at different time points post injury (*Figure 3*). For direct comparison, we utilized isolectin-B4 (IB4) and a myeloid-specific peroxidase (Mpx) antibody to label macrophages and neutrophils respectively in both organisms. IB4 labels macrophages in embryonic mouse tissues and adult hearts post infarction (*Maddox et al., 1982*; *Sorokin and Hoyt, 1992*; *Sorokin et al., 1994*; *Ismail et al., 2003*). In order to test the specificity of IB4 labeling in adult zebrafish hearts, we stained adult heart sections of validated macrophage reporter lines, *Tg(mpeg1.4:mCherry-F)* and *Tg (mpeg1:EGFP)* (*Figure 4—figure supplement 1.* and Figure 5D). Our data indicate that IB4+ cells largely overlap with *mpeg+* cells (*Figure 4—figure supplement 1* and Figure 5D). On the other hand, strong expression of Mpx, as well as its peroxidase activity, are features widely used to mark neutrophils (*Lieschke et al., 2001*; *Bennett et al., 2001*; *Renshaw et al., 2006*). To test the specificity of the Mpx antibody, we performed immunostaining on *TgBAC(mpx:GFP)* zebrafish hearts and found that the Mpx antibody exclusively labels *mpx*:GFP+ cells (*Figure 3—figure supplement 1*). Similarly, IB4 and the Mpx antibody label distinct immune cell-like populations in medaka hearts (*Figure 3A* and Figure 6A).

In the injured zebrafish heart, both macrophages and neutrophils were detected at 6 hpci (*Figure 3A*, quantification in 3B and *Figure 3—source data 1*). Neutrophils were more abundant than macrophages at this stage, consistent with observations in the fin amputation model (*Li et al., 2012*). Interestingly, we observed a peak in neutrophil recruitment at 2 dpci and a clear reduction by 7 dpci, indicating that neutrophil clearance occurs between 2 and 7 dpci in zebrafish (*Figure 3B*). On

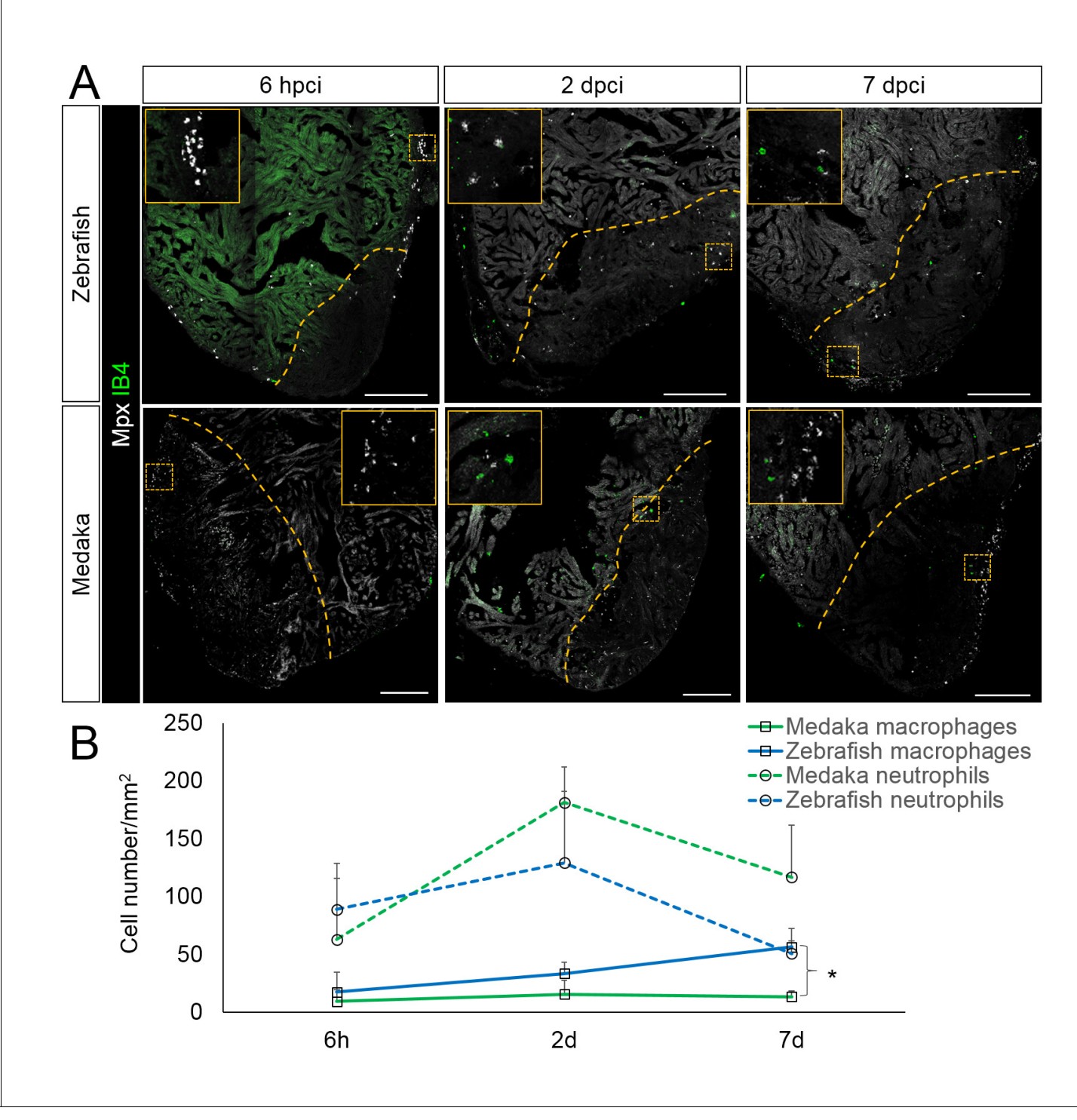

**Figure 3.** Immune cells dynamics in cryoinjured hearts. (**A**) Zebrafish and medaka heart sections at 6 hpci, 2 dpci and 7 dpci were stained with isolectin-B4 (IB4) for macrophages, and Mpx antibody for neutrophils. Positive cells, both in the injured area itself and within 100 µm of the injured area, were quantified (B; n = 3). Dotted lines delineate the injured area; scale bars, 200 µm. Macrophage numbers in medaka were always lower than those in zebrafish and this difference was especially pronounced at 7 pd. Neutrophil numbers appeared similar at 6 hpci, were higher in medaka at 2 dpci, and their clearance was delayed in medaka compared to zebrafish.

The following source data and figure supplements are available for figure 3:

**Source data 1.** Quantification of immune cells in cryoinjured hearts.

*Figure 3 continued on next page*

*Figure 3 continued*

**Figure supplement 1.** Mpx antibody specifically stains *mpx*:GFP-expressing cells.

**Figure supplement 1—source data 1.** Quantification of Mpx+ and *mpx*:GFP+ cells in zebrafish hearts following cardiac injury.

the other hand, macrophage numbers in zebrafish were initially low at 6 hpci and increased within the injured area by 7 dpci. However, when compared to zebrafish, macrophage recruitment in medaka was slower and significantly reduced at 7 dpci (*Figure 3B*). Furthermore, while neutrophil recruitment in medaka at 6 hpci appeared similar to zebrafish and also peaked at 2 dpci, their numbers were higher at 2 dpci and clearance was not as efficient by 7 dpci (*Figure 3B*). Prolonged neutrophils in the injured area might contribute to the excessive fibrotic response observed in medaka hearts post injury (*Ito et al., 2014*). In summary, our data indicate that compared to zebrafish, macrophage recruitment and neutrophil clearance in medaka are delayed following cardiac injury.

## Delayed macrophage recruitment in zebrafish compromises neovascularization, neutrophil clearance, CM proliferation, and scar resolution

To test the importance of timely macrophage recruitment in promoting heart regeneration in zebrafish, we delayed macrophage recruitment by pre-depletion using clodronate liposomes (CL) injections one day prior to injury. Administration of CL is an established method for macrophage depletion in mice, salamanders, and zebrafish (*Godwin et al., 2013*; *Aurora et al., 2014*; *de Preux Charles et al., 2016a*). Indeed, while macrophages (both *mpeg1*:EGFP+ and IB4+) in control hearts were clearly present in the injured area at 1 dpci, this recruitment was significantly reduced in CL-exposed hearts, and we observed injured areas devoid of macrophages at 1 dpci (*Figure 4A*, quantification in *Figure 4—figure supplement 1* and *Figure 4—figure supplement 1—source data 1*). On the other hand, neutrophil numbers were similar in CL-exposed and control hearts (quantification in *Figure 4—figure supplement 1* and *Figure 4—figure supplement 1—source data 1*). By 7 dpci, an increase in macrophage number was observed in CL-exposed hearts compared to controls, indicating that the treatment only delayed macrophage recruitment and invasion into the injured area (*Figure 4B*, quantification in *Figure 4—figure supplement 1* and *Figure 4—figure supplement 1—source data 1*).

When we examined neovascularization in these hearts, we observed a clear reduction in vessel formation in CL-exposed hearts compared to controls at 7 dpci, although angiogenic sprouting was not completely blocked (*Figure 4B*). Coincident with reduced neovascularization, CM proliferation was also significantly reduced in CL-exposed hearts compared to controls at 7 dpci (*Figure 4C*, quantification in 4D and *Figure 4—source data 1*). Of note, it has been reported that CL treatment does not directly depress CM proliferation in adult zebrafish hearts (*de Preux Charles et al., 2016a*), while the potential effect of CL on endothelial cells, or other cardiac cell types, has not been investigated to our knowledge. Although a few vessels could be observed in CL-exposed hearts at 7 dpci (*Figure 4B*), the scar tissue was completely avascular and unresolved by 1 mpci, suggesting that the few vessels observed at 7 dpci were not sufficiently stable to revascularize the injured area. Furthermore, Acid Fuchsin Orange G (AFOG) staining revealed that although the scar tissue in CL-exposed hearts had a fibrin and collagen composition similar to control, it was significantly larger and unresolved at 1mpci (*Figure 4F* and *Figure 4—figure supplement 2*, quantification in 4G and *Figure 4—source data 2*). This observation was in stark contrast to control hearts, where the coronary network was reestablished and minimal scarring was observed (*Figure 4E*).

Lastly, we observed that neutrophil clearance was compromised in CL-exposed hearts (*Figure 5*). Examination of control whole mount *TgBAC(mpx:GFP)$^{i114}$* hearts led to similar observations as described previously (*Figure 3*); neutrophil recruitment occurred within hours following injury, peaked at 2 dpci, and was followed by a decrease in neutrophil number and sporadic distribution in the injured area at 7 dpci (*Figure 5A*, quantification in *Figure 4—figure supplement 1* and *Figure 4—figure supplement 1—source data 1*). By 7 dpci, macrophage clearance of neutrophils was

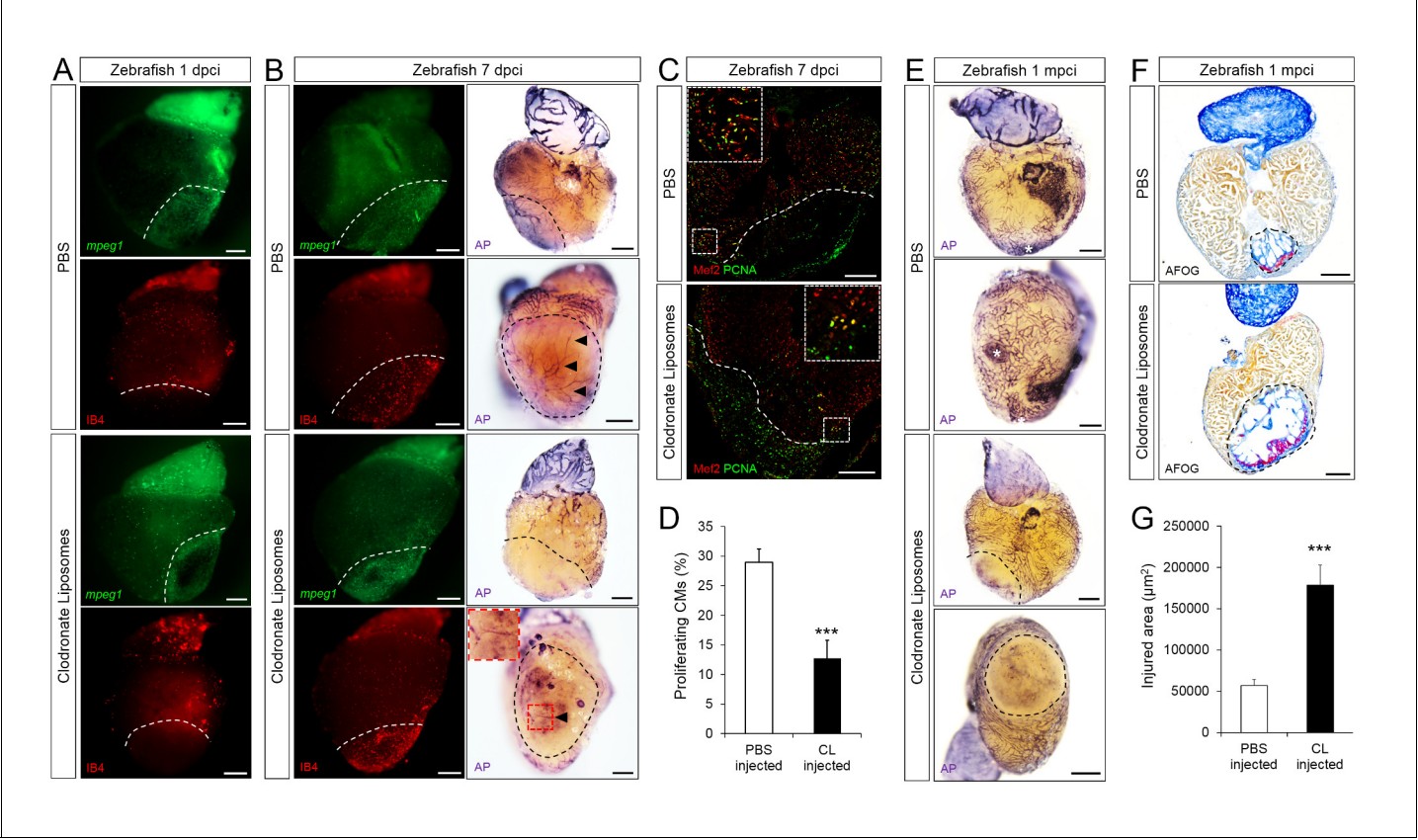

**Figure 4.** Delayed macrophage recruitment in zebrafish compromises neovascularization, CM proliferation and scar resolution. Adult zebrafish were injected with PBS or clodronate liposomes 1 day before injury. Hearts were collected at 1 dpci (**A**), 7 dpci (**B** and **C**) and 1 mpci (**E–G**) to examine macrophages by *mpeg1*:EGFP expression and IB4 staining (**A** and **B**), neovascularization by alkaline phosphatase (AP) staining (**B** and **E**, a side view as well as an apex view are shown), CM proliferation by PCNA/Mef2 immunostaining (**C**), and scar resolution by Acid Fuchsin Orange G (AFOG) staining (**F**). Mef2 and PCNA double positive CMs within 200 μm of the injured area were quantified in (**D**). Sections with the largest injured area/scar for each 1 mpci heart (delineated by black dotted lines) were stained with AFOG and quantified in G (n > 5, shown in *Figure 4—figure supplement 2*). Dotted lines delineate the injured area, arrowheads point to vessels, asterisks mark the initial injury site; scale bars, 200 μm. N ≥ 3 for each treatment and time point. Clodronate injections diminished macrophage recruitment at 1 dpci, and macrophage numbers recovered to control levels at 7 dpci (A and B, quantification in *Figure 4—figure supplement 1* and *Figure 4—figure supplement 1—source data 1*). Delayed macrophage recruitment compromised neovascularization (B and E), and CM proliferation (C and D), and resulted in delayed scar resolution (F and G).

The following source data and figure supplements are available for figure 4:

**Source data 1.** Quantification of proliferating CMs in control and CL-exposed zebrafish hearts following cardiac injury.

**Source data 2.** Quantification of scar areas in control and CL-exposed zebrafish hearts following cardiac injury.

**Figure supplement 1.** Delayed macrophage recruitment in zebrafish by Clodronate liposome pre-depletion.

**Figure supplement 1—source data 1.** Quantification of macrophage and neutrophil numbers in control and CL-exposed zebrafish hearts following cardiac injury.

**Figure supplement 2.** Delayed macrophage recruitment in zebrafish compromises scar resolution.

also observed, as indicated by inclusion of Mpx+ structures in IB4+ macrophages (*Figure 5B*), suggesting that neutrophils infiltrating the injured area are at least partly cleared by macrophage phagocytosis. On the other hand, neutrophil recruitment to the injured area seemed unaffected in CL-exposed hearts at 6 hpci, and continuous recruitment, possibly through coronaries was still

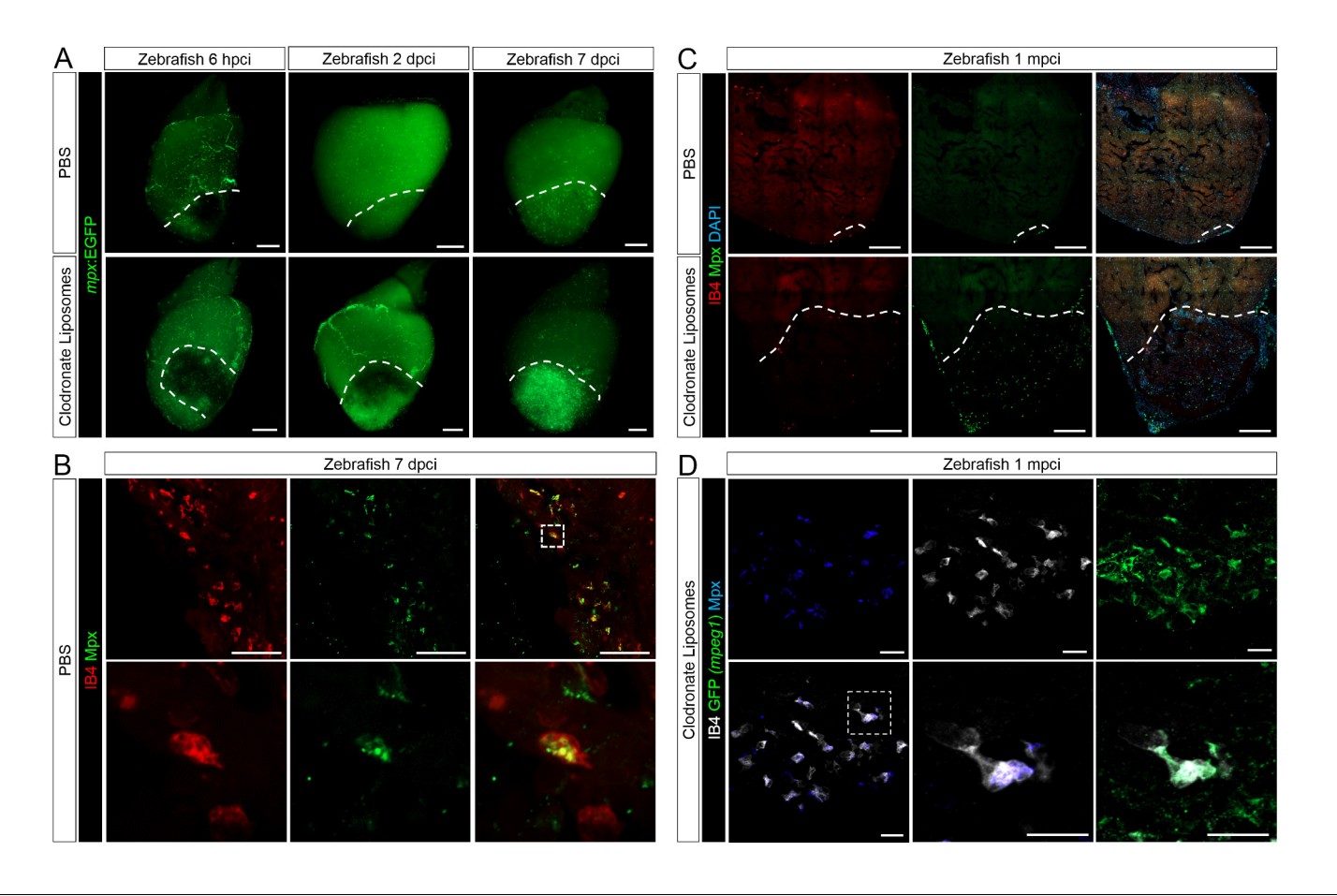

**Figure 5.** Neutrophil clearance is delayed after macrophage pre-depletion. Adult zebrafish were injected with PBS or clodronate liposomes 1 day before injury. Hearts were collected at 6 hpci (**A**), 2 dpci (**A**), 7 dpci (**A** and **B**), and 1 mpci (**C** and **D**) to examine macrophages by both IB4 staining and *mpeg1*:EGFP expression, and neutrophils by immunostaining for both Mpx and *mpx*:GFP expression. (**A**) Neutrophil dynamics at 6 hpci, 2 and 7 dpci. (**B**) Neutrophil clearance by macrophage phagocytosis was observed in PBS-exposed hearts at 7 dpci. (**C**) Neutrophils remained in the injured area of CL-exposed hearts at 1 mpci. (**D**) Neutrophil clearance by IB4- and *mpeg1*:EGFP-double positive macrophages was observed in CL-exposed hearts at 1 mpci. Dotted lines delineate injured area; scale bars, 200 μm (**A**), 50 μm (**B**), 100 μm (**C**), and 20 μm (**D**).

evident at 2 dpci (*Figure 5A*). However, instead of clearing from 2 to 7 dpci as in control hearts, neutrophil numbers in the injured area were significantly higher in CL-exposed hearts than in control hearts at 7 dpci (*Figure 5A*, quantification in *Figure 4—figure supplement 1* and *Figure 4—figure supplement 1—source data 1*), and neutrophils were still abundant in CL-exposed hearts at 1 mpci (*Figure 5C*). Consistently, macrophage-mediated neutrophil clearance was not observed at 7 dpci in CL-exposed hearts as in controls, but it was observed at 1 mpci (*Figure 5D*). These results suggest that timely macrophage recruitment is essential for coordinating neovascularization, neutrophil clearance, CM proliferation, and scar resolution during heart regeneration in zebrafish.

## Poly I:C injection in medaka promotes macrophage recruitment, neovascularization, CM proliferation, and scar resolution

From our transcriptomic analyses, the TLR agonist poly I:C was predicted to be an upstream regulator of cardiac repair in zebrafish when compared to medaka. TLRs have previously been shown to respond to tissue damage, initiating inflammation and immune cell recruitment (*Goulopoulou et al., 2016*). Thus, we tested whether poly I:C, and presumed activation of TLR signaling, could promote heart regeneration in the normally non-regenerative medaka. We performed intraperitoneal

injections of poly I:C immediately after cardiac injury, and subsequently examined macrophage and neutrophil recruitment in the medaka heart. We found that indeed, poly I:C injections significantly accelerated macrophage recruitment to the injured heart compared to control (*Figure 6A*, quantification in 6B and *Figure 6—source data 1*). Moreover, the number of neutrophils at the injured area after poly I:C injections was also reduced at 7 dpci when compared to control (*Figure 6*). These results suggest that poly I:C injections promote macrophage recruitment and neutrophil clearance in medaka.

We then examined whether poly I:C injections, and consequent modulation of immune cell dynamics, affected neovascularization, CM proliferation and scar resolution in injured medaka hearts. Consistent with previous data (*Ito et al., 2014*), we did not observe any vessels in the injured area of control medaka hearts before 7 dpci (*Figure 7A–C and and A'-C'*). At 14 dpci in control medaka hearts, we observed new vessel-like structures connecting to the preexisting plexus (*Figure 7D and D'*), although these vessel-like structures appeared unstable and had largely disappeared by 1 mpci (*Figure 7E and E'*). Surprisingly, in poly I:C-exposed hearts, we observed random vessel-like structures from 4 dpci onward (*Figure 7F–G and and F'-G'*). These vessel-like structures connected to each other to form a plexus at 7 dpci (*Figure 7H and H'*), increased in density at 14 dpci (*Figure 7I and I'*), and were still present at 1 mpci (*Figure 7J and J'*). Furthermore, combining macrophage pre-depletion by CL with poly I:C injections led to a block in early vessel formation (*Figure 7K and K'*). These results suggest that poly I:C promotes neovascularization, possibly in a macrophage-dependent manner.

Coincident with improved neovascularization, CM proliferation at 7 dpci was significantly increased in poly I:C-exposed hearts compared to controls (*Figure 8A*, and quantification in *Figure 8B* and *Figure 8—source data 1*). And this effect was blocked when we pre-depleted

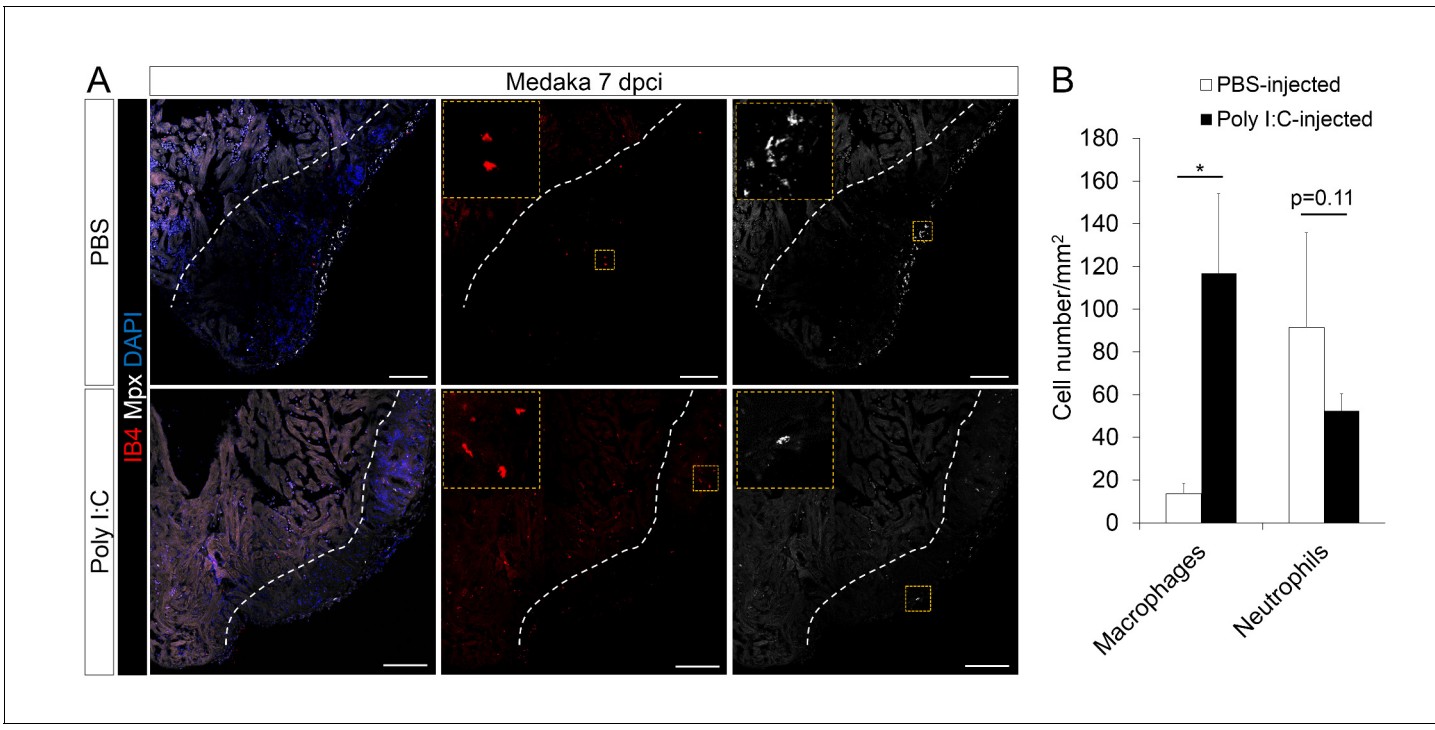

**Figure 6.** Poly I:C injection in medaka promotes macrophage recruitment and neutrophil clearance following cardiac injury. (**A**) Medaka heart sections at 7 dpci were stained with IB4 for macrophages and Mpx antibody for neutrophils. Positive cells, both in the injured area itself and within 100 μm of the injured area, were quantified (**B**; n = 3). Dotted lines delineate the injured area; scale bars, 100 μm. Poly I:C injection significantly promoted macrophage recruitment and neutrophil clearance in cryoinjured medaka hearts at 7 dpci.

The following source data is available for figure 6:

**Source data 1.** Quantification of macrophage and neutrophil numbers in control and poly I:C-exposed medaka hearts following cardiac injury.

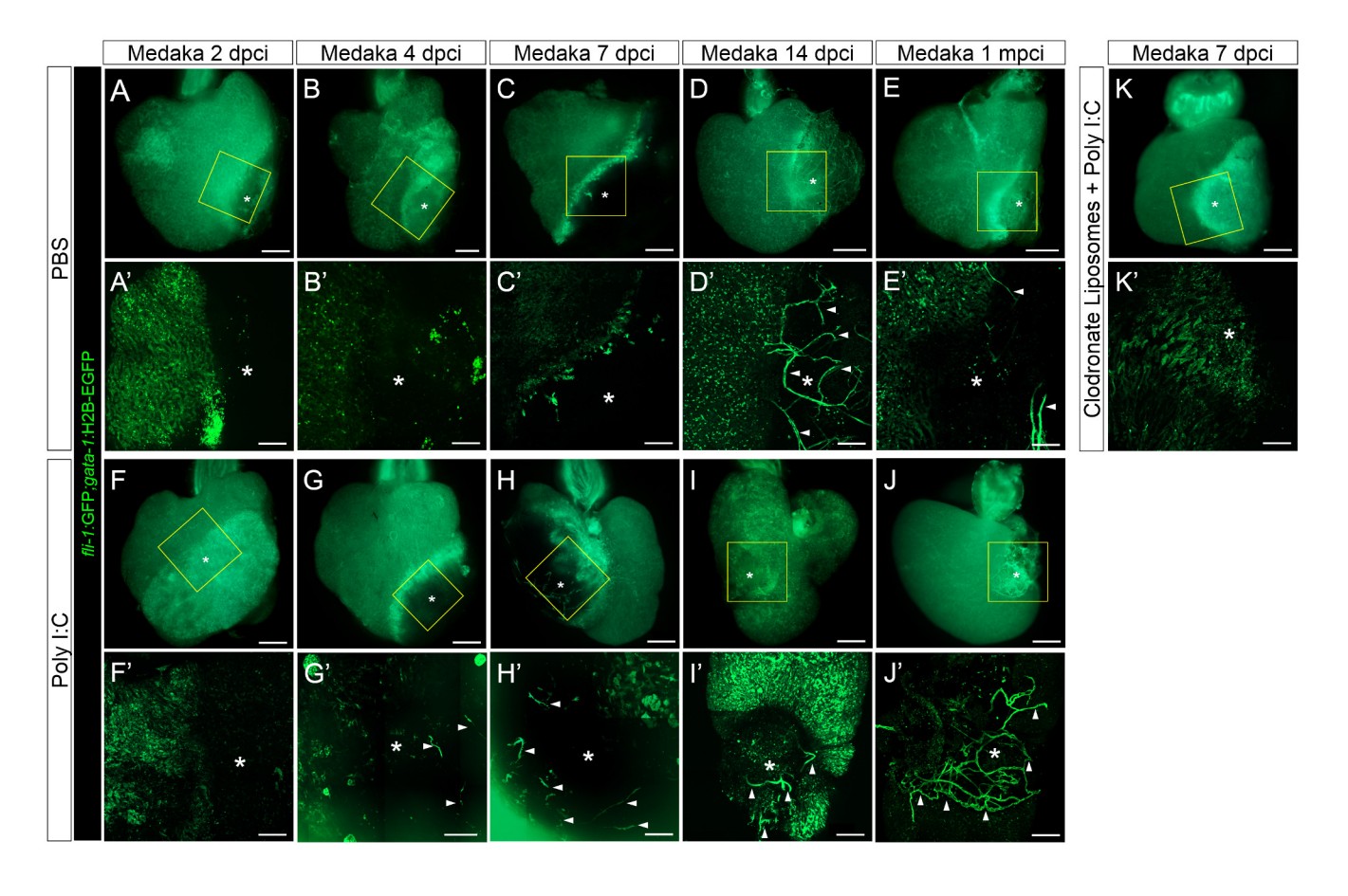

**Figure 7.** Poly I:C injection in medaka promotes neovascularization in a macrophage-dependent manner following cardiac injury. Adult *fli::GFP;gata1:: GFP* medaka were injected with PBS (A–E) or poly I:C (F–J) immediately after injury, or with clodronate liposomes at 1 day before injury and poly I:C immediately after injury (K). Hearts were collected at 2 (A and F), 4 (B and G), 7 (C, H, and K), and 14 (D and I) dpci, as well as 1 mpci (E and J), and imaged with a stereo (A–K) or confocal (A'-K', boxed areas in A-K) microscope for detailed examination. Asterisks mark the injured area, and arrowheads point to new vessel-like structures. Initiation of vessel formation was observed at earlier time points (4 and 7 dpci) in poly I:C-exposed hearts compared to control hearts, and these vessels were maintained at least until 1 mpci (n > 3). In contrast, in control hearts, new vessels were observed only at 14 dpci, and largely disappeared by 1 mpci. In addition, vessel formation promoted by poly I:C injections was blocked by CL pre-injections.

macrophages using CL injections, suggesting that poly I:C promotes CM proliferation in a macrophage-dependent manner. As a result of these regenerative responses, poly I:C-exposed hearts at 1 mpci showed significantly reduced scar tissue compared to controls (*Figure 9A*, quantification in *Figure 9B* and *Figure 9—source data 1*). Taken together, these data suggest that harnessing the acute immune response through TLR signaling activation can promote heart regeneration by accelerating macrophage recruitment, neutrophil clearance, neovascularization, CM proliferation and scar resolution in a species that normally does not regenerate its heart.

## Discussion

### Comparative transcriptomic analyses of zebrafish and medaka hearts post injury

Although comparative analyses have been performed in many species with the goal of identifying factors that may promote cardiac regeneration in humans (*Vivien et al., 2016*), direct comparison of

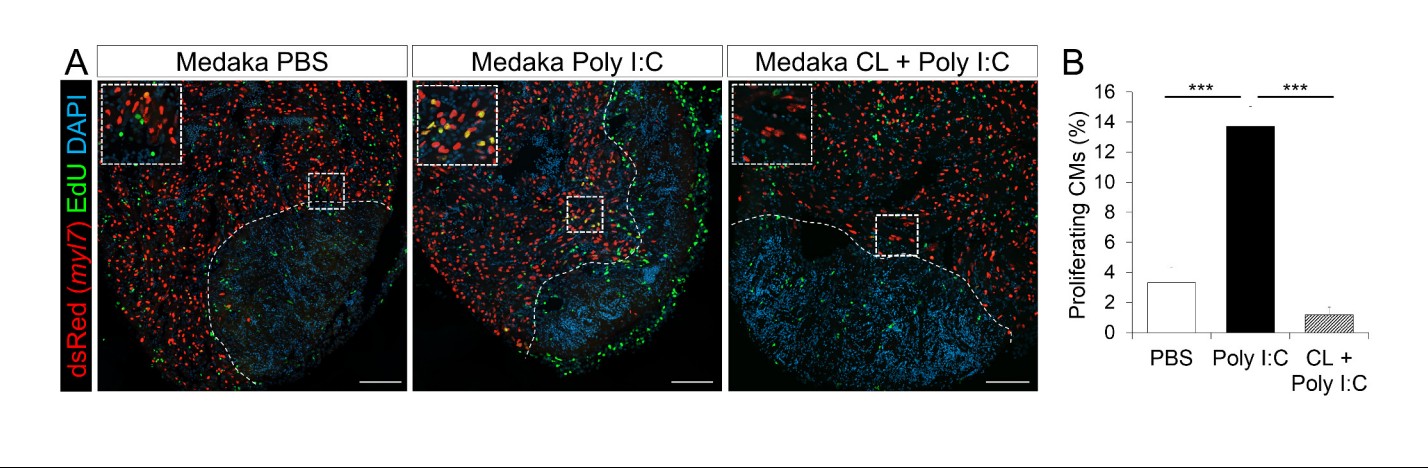

**Figure 8.** Poly I:C injection in medaka promotes CM proliferation in a macrophage-dependent manner following cardiac injury. Adult *zfmlc2 5.1 k: DsRED2-nuc* medaka were injected with PBS or poly I:C immediately after injury, or with clodronate liposomes at 1 day before injury and poly I:C immediately after injury. Hearts were collected at 7 dpci, and CM proliferation was examined by EdU labeling and immunostaining for DsRED expression; dotted lines delineate the injured area. EdU positive CMs within 200 µm of the injured area were quantified and shown (B; n ≥ 3). Scale bars, 100 µm. The percentage of proliferating CMs was significantly higher in poly I:C-exposed hearts compared to both PBS- and CL + poly I:C-exposed hearts.

The following source data is available for figure 8:

**Source data 1.** Quantification of proliferating CMs in medaka hearts following cardiac injury.

the transcriptome of species that are phylogenetically closely related has not been reported. Medaka share characteristics with species that can regenerate their hearts, including low-metabolic state, poikilothermia, hypoxia, an immature CM structure, and an immature immune system, yet are incapable of heart regeneration (*Ito et al., 2014*; *Vivien et al., 2016*). In particular, activation of epicardial and endocardial signals, neovascularization, CM proliferation, or scar resolution were not observed in medaka after cardiac resection (*Ito et al., 2014*). Here we report a comparative transcriptomic study of the responses to cardiac cryoinjury in zebrafish and medaka. Taking advantage of the well-annotated genomes of zebrafish and medaka, and a homology-based gene mapping technique, we generated a detailed dataset containing more than 15000 genes orthologous between these species, covering multiple time points in the first week following injury. Enrichment analyses based on this dataset revealed important biological processes and signaling pathways differentially regulated between zebrafish and medaka following cardiac injury. Previous attempts to compare responses in regenerative models, such as zebrafish, vs. non-regenerative models, such as adult mice, have been hindered by challenges to assign gene orthology, as well as basic physiological differences.

Here, we identified key processes such as the immune response, angiogenesis and cell proliferation that were delayed in medaka when compared to zebrafish. These data indicate that early responses to injury may play a key role in modulating subsequent events required for regeneration. Specifically, we observed that phagocytosis, as well as NF-kB, PI3K/AKT, NFAT, and TLR signaling pathways are blunted in their response following cardiac injury in medaka when compared to zebrafish. The presence in our datasets of pathways and signaling molecules previously implicated in heart regeneration further supports the power of our strategy to identify key regulators. Altogether, these results provide new insights into mechanisms underlying cardiac regeneration, as well as a new platform to investigate new candidate genes and pathways.

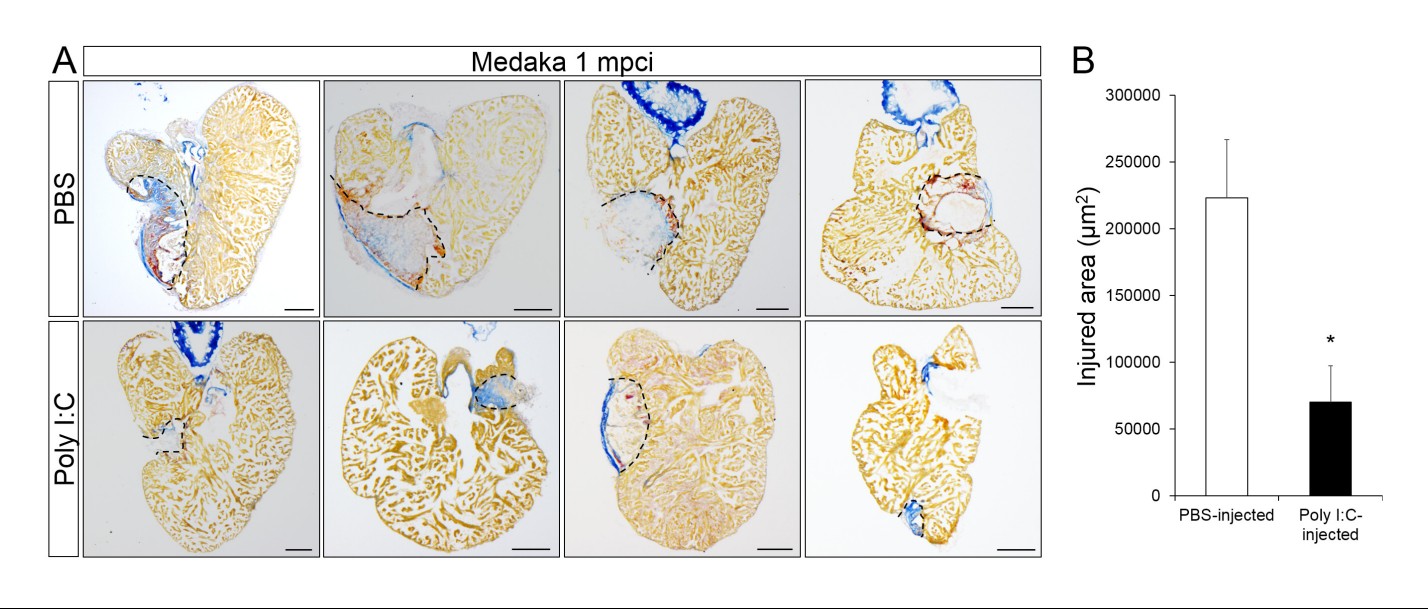

**Figure 9.** Poly I:C injection in medaka promotes scar resolution following cardiac injury. Adult *fli::GFP;gata1::GFP* medaka were injected with PBS or poly I:C immediately after injury. Hearts were collected at 1 mpci, and scar composition and resolution were examined by AFOG staining. (**A**) Healthy muscle stained in orange, fibrin in red, and collagen in blue. Sections with the largest scar area for each heart (delineated by dotted lines) were selected and quantified (**B**; n ≥ 4). Scale bars, 200 μm. Scar tissue in poly I:C-exposed hearts was significantly smaller than in controls at 1 mpci.

The following source data is available for figure 9:

**Source data 1.** Quantification of scar areas in medaka hearts following cardiac injury.

## Timely macrophage recruitment following injury is crucial for heart regeneration

Recently, we reported that fast angiogenic revascularization is a key process during heart regeneration (*Marín-Juez et al., 2016*), underscoring the importance of an efficient coordination of the early events following injury. In that study, angiogenic sprouting was blocked by overexpressing *dn-vegfaa* in the first days following injury, and it failed to occur subsequently (*Marín-Juez et al., 2016*). Here, we observed that a transient delay in macrophage recruitment at the time of injury also compromised neovascularization and heart regeneration. Previous studies investigating the role of macrophages during heart regeneration were performed by prolonged administration of CL (*Aurora et al., 2014*; *de Preux Charles et al., 2016a*), potentially interfering with the many different processes taking place during injury resolution. To assess the importance of the precise synchronization of the early events following injury, we found that early interference with macrophage recruitment was sufficient to disrupt neovascularization, neutrophil clearance, and heart regeneration. In a mouse MI model, CMs secrete REG3ß to recruit macrophages post injury, and the loss of REG3$\beta$ causes a large decrease in macrophage numbers in the ischemic heart, accompanied by insufficient clearance of neutrophils and increased ventricular dilatation (*Lörchner et al., 2015*). Similarly, we observed delayed and reduced macrophage recruitment as well as delayed neutrophil clearance in medaka hearts following injury. Altogether these observations from mice, zebrafish and medaka indicate that timely macrophage recruitment is required for cardiac regeneration.

## Poly I:C injection in medaka promotes heart regeneration

Delayed macrophage recruitment in zebrafish compromises heart regeneration, and macrophage recruitment is initiated and propagated at least in part through the recognition of DAMPs by TLRs (*Grote et al., 2011*; *Goulopoulou et al., 2016*). Thus, we tested whether enhanced TLR signaling via poly I:C injections would promote macrophage recruitment and heart regeneration in the non-regenerative medaka. Neovascularization of the injured area in zebrafish is one of the first processes

that occurs following injury, accomplished by angiogenic sprouting from existing coronaries, with potential contributions from epicardial-derived cells, and under the regulation of a group of growth factors including FGF, PDGF and VEGF (*Kim et al., 2010*; *Kikuchi et al., 2011*; *Marín-Juez et al., 2016*). Unlike zebrafish, medaka do not have distinct coronaries, and, instead, develop a coronary-like structure with interdigitating endocardial extensions that penetrate the compact myocardium (healthy side in *Figure 7A'*) (*Lemanski et al., 1975*; *Grimes et al., 2010*). This observation led to the suggestion that following cardiac injury, medaka lack neovascularization from epicardium-derived or other cell populations (*Lemanski et al., 1975*; *Ito et al., 2014*). In contrast, we show here that medaka heart grows new vessel-like structures that have connected to the preexisting plexus in the injured area by 14 dpci (*Figure 7D and D'*), a process that occurs much more slowly than the corresponding one in zebrafish (*Marín-Juez et al., 2016*). Still, these vessels appear to be unstable and are incapable of supporting regeneration.

Strikingly, we found that injections of poly I:C, a mimic of dsRNA and a TLR agonist, promote macrophage recruitment, faster and more stable neovascularization, increased CM proliferation, and scar resolution in medaka. TLR-mediated angiogenesis (or inflammation-induced angiogenesis) has been described in chronic inflammatory disorders, cardiovascular diseases, and cancer (*Grote et al., 2011*). TLR signaling could potentially trigger angiogenesis in two different ways. TLR activation in macrophages could stimulate VEGF expression and secretion, and consequently induce endothelial cell migration and angiogenesis (*Meda et al., 2012*). Similarly, TLR activation in endothelial cells could stimulate VEGF expression and cell proliferation, migration and tube formation (*Hu et al., 2016*). In addition, inflammatory signals have been proposed to prime CMs to dedifferentiate and proliferate in zebrafish (*de Preux Charles et al., 2016b*) as well as in neonatal mice (*Han et al., 2015*). In this study, we found that promotion of vessel formation and CM proliferation by poly I:C injections seem to be mediated by macrophages, as pre-depletion of macrophages by CL injections blocked these processes. Nevertheless, it is likely that poly I:C has additional effects.

In summary, our results reveal the relationship between timely macrophage recruitment, neovascularization, and neutrophil clearance in injured hearts, and highlight the importance of acute immune activation in coordinating the early events during cardiac regeneration. Thus, harnessing the immune response through TLR signaling, or otherwise, could be part of therapeutic interventions to improve tissue repair post myocardial infarction or other injuries.

## Materials and methods

### Zebrafish and medaka strains

All zebrafish and medaka husbandry was performed under standard conditions, and all animal experiments were done in accordance with institutional (MPG) and national ethical and animal welfare guidelines approved by the ethics committee for animal experiments at the Regierungspräsidium Darmstadt, Germany (permit numbers B2-1023 and B2-1111). We used the zebrafish AB strain (RRID:ZIRC_ZL1) as well as the following transgenic lines *Tg(mpeg1:EGFP)$^{gl22}$* (*Ellett et al., 2011*), *Tg(mpeg1.4:mCherry-F)$^{ump2}$* (*Bernut et al., 2014*), and *TgBAC(mpx:GFP)$^{i114}$* (*Renshaw et al., 2006*). We used the medaka WT Cab strain as well as the following transgenic lines *fli::GFP;gata1::GFP* (*Schaafhausen et al., 2013*), *fli1-GFP* (*Moriyama et al., 2010*; *Ito et al., 2014*), and *Cab-Tg(zfmlc2 5.1 k:DsRed2-nuc)$^{TG1026}$* (*Taneda et al., 2010*). The two medaka *fli* alleles were generated independently using the same plasmid, and we did not observe any difference in expression between them. All animals were maintained at 28°C in a recirculating system and fish >4 months of age were used in all experiments. Intraperitoneal (IP) injections were performed as previously described (*Kinkel et al., 2010*). PBS or Clodronate liposome (5 mg/ml, ClodronateLiposomes.org) was injected at 10 µl per zebrafish or medaka one day before cryoinjury, and PBS or poly I:C (1 µg/µl, R&D Systems, Minneapolis, MN) was injected at 10 µl per fish immediately after cryoinjury. Cryoinjury was performed as previously described in zebrafish (*González-Rosa et al., 2011*; *Chablais et al., 2011*, *Schnabel et al., 2011*), and the same method was used in medaka.

### RNA sequencing and primary analyses

For RNA-seq, RNA was isolated from zebrafish and medaka ventricles using the miRNeasy micro Kit (Qiagen, Valencia, CA) combined with on-column DNase digestion (DNase-Free DNase Set, Qiagen)

to avoid contamination by genomic DNA. RNA and library preparation integrity were verified with a BioAnalyzer 2100 (Agilent Technologies, Santa Clara, CA) or LabChip Gx Touch 24 (Perkin-Elmer, Nieuwerkerk a.d. IJsel, The Netherlands). 500 ng of total RNA was used as input for Truseq Stranded mRNA Library preparation following the low sample protocol (Illumina Inc. San Diego, CA). Sequencing was performed on a NextSeq500 instrument (Illumina) using v1 chemistry, resulting in a minimum of 20–30M reads per library with a $2 \times 75$ bp paired end setup. The resulting raw reads were assessed for quality, adapter content and duplication rates with FastQC (Andrews S. 2010, FastQC: a quality control tool for high throughput sequence data. Available online at: http://www.bioinformatics.babraham.ac.uk/projects/fastqc). Trimmomatic version 0.33 was employed to trim reads after a quality drop below a mean of Q18 in a window of 5 nucleotides (*Bolger et al., 2014*). Only reads above 30 nucleotides were cleared for further analyses. Trimmed and filtered reads were aligned versus the Ensembl release 80 of zebrafish assembly 10 (*Danio rerio* GRCz10) and Ensembl release 80 of medaka assembly 1 (*Oryzias latipes* MEDAKA1) using STAR 2.4.0a with the parameter '–outFilterMismatchNoverLmax 0.1' to increase the maximum ratio of mismatches to mapped length to 10% (*Dobin et al., 2013*). The number of reads aligning to genes was counted with the featureCounts 1.4.5-p1 tool from the Subread package (*Liao et al., 2014*). Only reads mapping at least partially inside exons were admitted and aggregated per gene. Reads overlapping multiple genes or aligning to multiple regions were excluded. The Ensemble annotation was enriched with UniProt data (release 06.06.2014) based on Ensembl gene identifiers (Activities at the Universal Protein Resource, UniProt). A comparison based on Biomart (http://www.ensembl.org/biomart) revealed 15292 genes orthologous between zebrafish and medaka showing a minimum sum of 5 reads among all time points. Raw counts of these 15292 orthologous genes were joined to a combined matrix and normalized using DESeq. Further analyses are based on these values and the respective $\log_2$ transformed fold changes resulting from specified contrasts of cryoinjured versus untouched fish.

## Gene Ontology enrichment analysis

All genes with an absolute $\log_2$ fold change >2 against the respective 0 hr (untouched) sample in at least one condition were used for Gene Ontology overrepresentation/underrepresentation analysis with the cytoscape BINGO plugin. Hypergeometric test and Benjamini and Hochberg FDR correction were conducted with a significance level of 0.05, while the whole annotation set was used as the background set.

## Principal component analysis

Principal component analysis (PCA) was performed on the DESeq normalized counts of untouched and cryoinjured zebrafish and medaka RNAseq gene expression datasets to analyze the general sample progression through time by partitioning variability of the data.

## Sample level enrichment analysis (SLEA)

In preparation for pathway enrichment analyses, $\log_2$ transformed fold changes resulting from contrasts of cryoinjured versus untouched fish (columns) were computed for all available genes (rows) having at least 5 DESeq normalized expression counts per time point. Fold changes of each contrast were linearly scaled between $-10$ and $+10$ and centered by subtraction of the column mean from each value to normalize for differing distributions. The resulting fold change matrix was imported into the Gitools framework 2.2.2 for SLEA analyses (http://www.gitools.org) (*Perez-Llamas and Lopez-Bigas, 2011*). Modules for gene annotations, GO terms and KEGG pathways were downloaded from the Ensembl database (http://www.ensembl.org). Enriched modules were subjected to statistical analysis by the Z-score method, to compare the mean expression value of each module to a distribution of the mean of 10,000 random modules of the same size created with the expression values of the same sample. The result of this analysis is a Z-score, which measures the difference between observed and expected mean expression values for a gene set. A module was defined as statistically significant above a Z-score of $\pm 4$. Results were visualized as heat maps, with red denoting activation and blue denoting inactivation.

## Differential expression of untouched and cryoinjured fish

The dataset used as input for SLEA was further filtered for genes showing differential expression (absolute $\log_2$ fold change $\geq 2$) in at least one contrast (cryoinjured/untouched for each time point) resulting in 5312 DEGs. These 5312 DEGs were visualized with heatmap2 function in R.

## Pathway analysis

Pathway analysis was performed using Ingenuity Pathway Analysis (IPA) software (Qiagen) following manufacturer's instructions. We input data from the normalized orthologous gene list between zebrafish and medaka, and defined DEGs as $\log_2$ FC >1. Canonical pathways and upstream regulators were presented in Z-scores.

## Cryosection and histological analyses

Zebrafish and medaka hearts were extracted and fixed in 4% (wt/vol) paraformaldehyde at room temperature for 1 hr and subsequently cryopreserved with 30% (wt/vol) sucrose before immersion in OCT (Tissue-Tek, Sakura Finetek, Torrance, CA) and immediately stored at $-80°C$. 12 µm cryosections were collected for histological analysis. AFOG staining was performed by using an AFOG staining kit (Gennova Scientific S.L. Seville, Spain) following manufacturer's instructions with the following changes: Samples were first incubated at 60°C for 2 hr with Bouin preheated at 60°C for 30 min; slides were then rinsed in running water for 30 min, and incubated with phosphomolybdic acid solution, and subsequent steps were carried out by following manufacturer's instructions. Quantification for each heart was done by measuring the scar area from the section with the largest one.

## Immunostaining and imaging

For immunofluorescence, samples were washed twice with PBST ($1\times$ PBS, 0.1% Triton X-100) and twice with $dH_2O$ before permeabilization with 3% (vol/vol) $H_2O_2$ in methanol for 1 hr at room temperature. Samples were then washed twice with PBST and incubated in blocking solution [$1\times$ PBS, 2% (vol/vol) sheep serum, 0.2% Triton X-100, 1% DMSO]. Samples were incubated in primary antibodies overnight at 4°C, followed by two PBST washes and incubation with secondary antibodies for 3 hr at room temperature. Samples were washed again with PBST, and DAPI (1:10000 in PBST) or SYTOX (1:500 in PBST, Thermo Fisher Scientific, Waltham, MA) was added before mounting. Antibodies and reagents used in this study include anti-GFP (Aves Labs, Portland, OR) at 1:500, anti-DsRED (Clontech, Mountain View, CA) at 1:200, anti-Mpx (Genetex, San Antonio, TX) at 1:200, Dylight 594-conjugated isolectin B4 (IB4-594, Vector Laboratories, Burlingame, CA) at 1:200, and Alexa Fluor 488 phalloidin (Thermo Fisher Scientific) at 1:500. EdU staining was performed by using the Click-iT EdU Imaging Kit (Molecular Probes, Eugene, OR) following manufacturer's instructions. EdU (250 µg/fish) was injected IP 24 hr before extraction of the heart. Alkaline phosphatase staining was performed as follows: hearts were fixed in 4% (wt/vol) paraformaldehyde for 1 hr at room temperature, washed three times with PBS for 10 min, equilibrated in alkaline buffer (100 mM Tris, pH 9.5, 100 mM NaCl, 0.1% Tween20), and then stained in NBT/BCIP stock solution (Roche Applied Science, Indianapolis, IN) 1:100 diluted in alkaline solution. The reaction was stopped by 3 PBS washes for 10 min each and samples were imaged as soon as proper vessel staining was revealed.

Imaging of whole-mount hearts and heart sections was performed by using Zeiss LSM 800 and Spinning Disk confocal microscopes, or a Nikon SMZ25. Quantification of cardiomyocyte proliferation in both zebrafish and medaka was performed in the 200 µm area directly adjacent to the injured area as previously described (*Marín-Juez et al., 2016*). Quantification of macrophage and neutrophil numbers was performed in the 100 µm area directly adjacent to, as well as within the injured area. To try and exclude other myeloid cells with potentially low *mpx* expression, we tested the specificity of our staining/imaging conditions in *TgBAC(mpx:GFP)[i114]* hearts, and set a cut-off to only count cells with substantial expression. The student's t-test was used to assess all comparisons.

## Acknowledgements

We thank Jochen Wittbrodt, Lazaro Centanin, Akira Kudo, and the NBRP Medaka resource center (https://shigen.nig.ac.jp/medaka/) for providing medaka lines, Hans-Martin Maischein for fish husbandry, Beate Grohmann, Radhan Ramadass and Rebecca I-Ching Lee for expert technical

assistance, Arica Beisaw, Yu Hsuan Carol Yang and Chi-Chung Wu for feedback on the manuscript, and all Stainier lab members for discussions.

## Additional information

### Competing interests

DYRS: Senior editor, *eLife*. The other authors declare that no competing interests exist.

### Funding

| Funder | Grant reference number | Author |
| --- | --- | --- |
| Max-Planck-Gesellschaft | Open-access funding | Didier YR Stainier |

The funder had no role in study design, data collection and interpretation, or the decision to submit the work for publication.

### Author contributions

S-LL, Conceptualization, Data curation, Formal analysis, Supervision, Validation, Investigation, Visualization, Writing—original draft, Writing—review and editing; RM-J, Data curation, Formal analysis, Investigation, Writing—review and editing; PLM, Data curation, Formal analysis, Investigation, Visualization, Writing—review and editing; CK, Data curation, Software, Formal analysis, Investigation, Methodology, Writing—review and editing; JKHL, Formal analysis, Investigation, Writing—review and editing; ATT, Investigation; SG, Data curation, Investigation; ML, Conceptualization, Data curation, Software, Formal analysis, Investigation, Writing—review and editing; DYRS, Conceptualization, Resources, Data curation, Supervision, Funding acquisition, Investigation, Project administration, Writing—review and editing

### Author ORCIDs

Shih-Lei Lai, http://orcid.org/0000-0002-1409-4701
Didier YR Stainier, http://orcid.org/0000-0002-0382-0026

### Ethics

Animal experimentation: All zebrafish and medaka husbandry was performed under standard conditions, and all animal experiments were done in accordance with institutional (MPG) and national ethical and animal welfare guidelines approved by the ethics committee for animal experiments at the Regierungspräsidium Darmstadt, Germany (permit numbers B2-1023 and B2-1111).

## Additional files

### Supplementary files

• Supplementary file 1. Normalized orthologous genes expression in zebrafish and medaka post cardiac injury.

• Supplementary file 2. List of biological processes revealed by Sample Level Enrichment Analysis.

• Supplementary file 3. Top canonical pathways and upstream regulators predicted by Ingenuity Pathway Analysis. (Z-zebrafish, M-medaka, H-hours post injury, D-days post injury.)

• Supplementary file 4. List of canonical pathways and upstream regulators predicted by Ingenuity Pathway Analysis.

### Major datasets

The following dataset was generated:

**Database, license,**

| Author(s) | Year | Dataset title | Dataset URL | and accessibility information |
|---|---|---|---|---|
| Lai S, Marín-Juez R, Moura P, Kuenne C, Lai KH, Tsedeke AT, Guenther S, Looso M, Stainier DY | 2017 | Comparative transcriptome profiling of zebrafish and medaka hearts following cardiac cryoinjury | https://www.ncbi.nlm.nih.gov/geo/query/acc.cgi?acc=GSE94617 | Publicly available at the NCBI Gene Expression Omnibus (accession no: GSE94617) |

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
