## [Decision Letter]

Thank you for submitting your article "Reciprocal analyses in zebrafish and medaka reveal that harnessing the immune response promotes cardiac regeneration" for consideration by *eLife*. Your article has been reviewed by three peer reviewers and the evaluation has been overseen by a Reviewing Editor and Marianne Bronner as the Senior Editor. The following individual involved in review of your submission has agreed to reveal their identity: Nadia Rosenthal (Reviewer #1).

The reviewers have discussed the reviews with one another and the Reviewing Editor has drafted this decision to help you prepare a revised submission.

Summary:

This is an interesting and important study in which the authors compare recovery from cryoinjury in zebrafish (adult cardiac regeneration capable) and medaka, another teleost fish (adult cardiac regeneration incompetent) using comparative transcriptomics. They describe immune cell recruitment and perform functional studies to deplete macrophages in zebrafish along with stimulation of regeneration with a TLR agonist in medaka. Their results provide evidence that innate immune cells are important in efficient cardiac regeneration, and that the timing and modulation of myeloid cells may have direct roles in regulating neovascularization, cardiomyocyte proliferation and scar resolution. While the requirement for macrophages in regeneration and clearance previously has been established, demonstrating this ability across species and testing limiting factors for regeneration is of broad interest.

Essential revisions:

1) Mpx protein is expressed in most myeloid cells in fish and amphibians. The Mpx promoter used in transgenic zebrafish is brighter in neutrophils than in other myeloid cells but depending on microscopy settings can be detectible in macrophages and monocytes (see Characterization of Zebrafish Larval Inflammatory Macrophages, Dev Comp Immunol. 2009, 33: 1212-1217). Similarly, although IB4 stains microglia and other macrophage populations in fish, not all macrophages may stain with the IB4 lectin or with Mpeg-1. This lack of specificity of these markers requires some caution when quantifying macrophages versus neutrophils in their models. This should be taken into consideration when interpreting results.

2) In Figure 2, although neutrophil chemotaxis-related genes appear to be upregulated in medaka relative to zebrafish, so do monocyte chemotaxis genes. This may be a sign that the situation is not as "cut and dry" in medaka as presented, and that monocytes rather than neutrophils may alter the inflammatory landscape. Notably, monocytes would be positive for Mpx. These confounders need to be taken into account when interpreting the neutrophil/ macrophage/ monocyte numbers with the tools in hand.

3) The study employs Poly I:C as a macrophage agonist operating via TLR3. While this mechanism is probably conserved in the cardiac context, the role for Poly I:C as a potent agonist for endothelial cells is probably also conserved (see Polyinosinic:polycytidylic acid is a potent activator of endothelial cells Am J Pathol. 1994, 145: 137-147). Poly I:C treatment in medaka results in transient endothelial activation and vessel growth, but it is not clear if this is a direct or indirect effect. If the authors seek to make this claim, studies should be performed using the existing Fli1 GFP line to clarify if Poly I:C is acting directly on the endothelial cell population or via myeloid cells in this model.

4) Of concern, Figure 5 is supposed to show reduced macrophage recruitment with clodronate treatment, however if the outflow tract is ignored it appears that at 1 dpci there are more Mpeg-1 positive cells present in the lesion of clodronate treated animals, while the number of IB4+ cells looks unchanged. At 7 dpci it is clear that there are more IB4 and Mpeg-1 positive cells in the lesion. This does not correlate with the description of the results in the text.

5) Figure 5 uses Mef2 to mark cardiomyocytes (CMs) whereas Figure 8/L reports enhanced CM proliferation using Edu with Phalloidin to mark CM's. How do the authors confidently assign EdU positive cells to CMs? Poly I:C is likely to stimulate proliferation of myeloid cells and endothelial cells in this model, so EdU staining should be performed in conjunction with a CM-specific marker to support their claim. Phalloidin also marks fibroblastic endothelial cells, and myofibroblasts which are both common in many wound environments. More specific markers should be used in this context such as Mef2, Nkx2.5 or sarcomeric myosin etc. to verify proliferation specifically of CMs

6) A major aspect of the paper that remains untested is whether the pro-regenerative role of macrophages is solely to reduce the neutrophil numbers. It would be meaningful for the authors to experimentally eliminate neutrophils in the medaka model and ask to what extent this promotes regeneration (in the absence or presence of macrophages). Alternatively, since there may be more NTR-type models available in zebrafish, eliminating neutrophils in chlodronate-treated zebrafish could also be used to determine the role of neutrophil clearance in promoting regeneration.

7) In Figure 5, it is not really evident that the chlodronate treatment eliminated any macrophages. It would be important to provide quantification as in Figure 4. The uncertainty about macrophage clearance raises the question whether the phenotype is truly due to the chlodronate-mediated effects on macrophages. In that sense, it is unclear if PBS is the appropriate control. PBS liposomes would be a better control for this experiment. Indeed a section in the Materials and methods on the chlodronate injections and the polyI:C seems to be missing, and is important.

[Editors' note: further revisions were requested prior to acceptance, as described below.]

Thank you for submitting your article "Reciprocal analyses in zebrafish and medaka reveal that harnessing the immune response promotes cardiac regeneration" for consideration by *eLife*. Your article has been reviewed by three peer reviewers and the evaluation has been overseen by a Reviewing Editor and Marianne Bronner as the Senior. The following individual involved in review of your submission has agreed to reveal their identity: Nadia Rosenthal (Reviewer #1).

While the revision is much improved and very close to being acceptable for publication, one of the reviewers has come up with an additional suggestion. Although we realize this was not raised in the original review, it was thought it would be a good control to treat with chlodronate during larval growth when cardiomyocytes are proliferative, to show that chlodronate does not depress normal cardiomyocyte proliferation. At a minimum, you should modify the text to make note that the specificity of the chlorinate treatment is unknown.

---

## [Author Response]

*Essential revisions:*

*1) Mpx protein is expressed in most myeloid cells in fish and amphibians. The Mpx promoter used in transgenic zebrafish is brighter in neutrophils than in other myeloid cells but depending on microscopy settings can be detectible in macrophages and monocytes (see Characterization of Zebrafish Larval Inflammatory Macrophages, Dev Comp Immunol. 2009, 33: 1212-1217). Similarly, although IB4 stains microglia and other macrophage populations in fish, not all macrophages may stain with the IB4 lectin or with Mpeg-1. This lack of specificity of these markers requires some caution when quantifying macrophages versus neutrophils in their models. This should be taken into consideration when interpreting results.*

We thank the reviewer for pointing out this issue. Indeed, as the reviewer indicated, some myeloid cell populations might display minimal levels of Mpx expression. Amongst the zebrafish *mpx* reporter lines, the one mentioned by the reviewer (Mathias et al. DCI 2009) was generated with only a portion of the *mpx* promoter (8 kb), which might be the cause for the expression the authors observed in macrophages. Since we were aware of these issues, we used only the *TgBAC(mpx:GFP)^i114^*line (Renshaw et al. Blood 2006) to maximize the specificity of the reagent.

This BAC transgenic line has been reported to specifically mark neutrophils in zebrafish; both l- plastin and neutral red stained monocytes/macrophages are negative for GFP in this line (Renshaw et al. 2006). We have also performed immunostaining for Mpx on *TgBAC(mpx:GFP)^i114^*samples to test the specificity, and show new data (Figure 4—figure supplement 1). The Mpx antibody immunostained 88.23% of the *mpx:*GFP+ cells, and with the settings used for all our analyses, we did not observe any Mpx staining of *mpx:*GFP- cells.

In addition, to test the specificity of IB4 staining for macrophages, we now show IB4 staining in *Tg(mpeg1.4:mCherry-F)^ump2^;TgBAC(mpx:GFP)^i114^*injured hearts at both 1 and 7 dpci (Figure 5—figure supplement 1), and in *Tg(mpeg1:EGFP)^gl22^*at 1 mpci at high resolution (Figure 6). These new data show that IB4 staining colocalizes with *mpeg1*:mCherry+ or *mpeg1*:EGFP+ cells, but not with *mpx*:GFP+ cells in all cases.

In medaka, we show mutually exclusive labeling of Mpx+ neutrophils and IB4+ macrophages in Figure 4,Figure 7. Since little is known about the different macrophage populations in the adult zebrafish, we now carefully indicate in the text whether the cell were IB4+ and/or *mpeg*+.

*2) In Figure 2, although neutrophil chemotaxis-related genes appear to be upregulated in medaka relative to zebrafish, so do monocyte chemotaxis genes. This may be a sign that the situation is not as "cut and dry" in medaka as presented, and that monocytes rather than neutrophils may alter the inflammatory landscape. Notably, monocytes would be positive for Mpx. These confounders need to be taken into account when interpreting the neutrophil/ macrophage/ monocyte numbers with the tools in hand.*

From the Gene Ontology website (http://www.geneontology.org/page/download-annotations), all 24 genes in the “monocyte chemotaxis” group are a subset of the 38 “neutrophil chemotaxis” genes. Thus, from a bioinformatics standpoint, we do not have sufficient evidence to distinguish one process from the other. Therefore, and as suggested by the reviewer, we have now carefully revised the manuscript in this regard (subsection “Macrophage recruitment in medaka heart post injury is delayed and reduced compared to zebrafish”).

*3) The study employs Poly I:C as a macrophage agonist operating via TLR3. While this mechanism is probably conserved in the cardiac context, the role for Poly I:C as a potent agonist for endothelial cells is probably also conserved (see Polyinosinic:polycytidylic acid is a potent activator of endothelial cells Am J Pathol. 1994, 145: 137-147). Poly I:C treatment in medaka results in transient endothelial activation and vessel growth, but it is not clear if this is a direct or indirect effect. If the authors seek to make this claim, studies should be performed using the existing Fli1 GFP line to clarify if Poly I:C is acting directly on the endothelial cell population or via myeloid cells in this model.*

To investigate whether the beneficial effects of poly I:C injections are mediated by macrophages or endothelial cells, we pre-depleted macrophages by clodronate liposomes (CL) injections 1 day prior to cardiac injury. We found that macrophage pre-depletion blocks both early vessel formation and CM proliferation at 7 dpci in poly I:C-exposed hearts (Figure 1,Figure 8). These new findings support the contribution of macrophages in the beneficial effects of poly I:C injections. Nevertheless, we cannot exclude the potential effects of poly I:C injections on endothelial/endocardial cells, and have modified the text accordingly (subsection “Poly I:C injection in medaka promotes heart regeneration”).

*4) Of concern, Figure 5 is supposed to show reduced macrophage recruitment with clodronate treatment, however if the outflow tract is ignored it appears that at 1 dpci there are more Mpeg-1 positive cells present in the lesion of clodronate treated animals, while the number of IB4+ cells looks unchanged. At 7 dpci it is clear that there are more IB4 and Mpeg-1 positive cells in the lesion. This does not correlate with the description of the results in the text.*

To address this concern, we have now quantified both neutrophils and macrophages in *Tg(mpeg1.4:mCherry-F)^ump2^;Tg(mpx:GFP)^i114^*heart sections at both 1 and 7 dpci (Figure 5—figure supplement 1), and observed that macrophages were significantly diminished in CL-exposed hearts at 1 dpci. By 7 dpci, macrophage numbers were comparable between control and CL-exposed hearts.

*5) Figure 5 uses Mef2 to mark cardiomyocytes (CMs) whereas Figure 8/L reports enhanced CM proliferation using Edu with Phalloidin to mark CM's. How do the authors confidently assign EdU positive cells to CMs? Poly I:C is likely to stimulate proliferation of myeloid cells and endothelial cells in this model, so EdU staining should be performed in conjunction with a CM-specific marker to support their claim. Phalloidin also marks fibroblastic endothelial cells, and myofibroblasts which are both common in many wound environments. More specific markers should be used in this context such as Mef2, Nkx2.5 or sarcomeric myosin etc. to verify proliferation specifically of CMs*

We agree with the reviewer’s suggestion and have now repeated the experiments using *Cab- Tg(zfmlc2 5.1k: DsRed2-nuc)^TG1026^*medaka, which express DsRed2 in the nuclei of their CMs. These new data show that poly I:C injections in medaka promote CM proliferation, which was blocked by pre-depletion of macrophages with CL injections (Figure 1).

*6) A major aspect of the paper that remains untested is whether the pro-regenerative role of macrophages is solely to reduce the neutrophil numbers. It would be meaningful for the authors to experimentally eliminate neutrophils in the medaka model and ask to what extent this promotes regeneration (in the absence or presence of macrophages). Alternatively, since there may be more NTR-type models available in zebrafish, eliminating neutrophils in chlodronate-treated zebrafish could also be used to determine the role of neutrophil clearance in promoting regeneration.*

Unfortunately and as pointed out by the reviewer, there is currently no neutrophil ablation medaka line. In order to ablate neutrophils in adult zebrafish, we crossed *Tg(mpx:Gal4)^SH267^*fish (Robertson et al., Sci Transl Med 2014) with *Tg(UAS-E1b:Eco.NfsB-mCherry)* fish. Although these reagents have been recently shown to efficiently deplete *mpx*+ cells during development (Matsuoka et al., *eLife*, 2016), we found that the *mpx:GAL4* transgene is silenced in adults, thus preventing neutrophil ablation experiments in adult fish. Therefore, new tools will need to be generated for these important experiments.

*7) In Figure 5, it is not really evident that the chlodronate treatment eliminated any macrophages. It would be important to provide quantification as in Figure 4. The uncertainty about macrophage clearance raises the question whether the phenotype is truly due to the chlodronate-mediated effects on macrophages. In that sense, it is unclear if PBS is the appropriate control. PBS liposomes would be a better control for this experiment. Indeed a section in the Materials and methods on the chlodronate injections and the polyI:C seems to be missing, and is important.*

As mentioned above (point #4), we have now quantified both neutrophils and macrophages in *Tg(mpeg1.4:mCherry-F)^ump2^;Tg(mpx:GFP)^i114^*heart sections at both 1 and 7 dpci (Figure 5—figure supplement 1), and observed that macrophages were significantly diminished in CL^-^exposed hearts at 1 dpci compared to control.

We also now provide more information about the CL and poly I:C injections in the Materials and methods section as suggested.